# New 2,3-Benzodiazepine Derivative: Synthesis, Activity on Central Nervous System, and Toxicity Study in Mice

**DOI:** 10.3390/ph14080814

**Published:** 2021-08-19

**Authors:** Amal Amaghnouje, Serhii Bohza, Nathalie Bohdan, Imane Es-Safi, Andrii Kyrylchuk, Sanae Achour, Hinde El Fatemi, Dalila Bousta, Andriy Grafov

**Affiliations:** 1Laboratory of Biotechnology, Environment, Agrofood and Health, Dhar al Mehraz Faculty of Sciences, Sidi Mohammed Ben Abdellah University, Fez 30003, Morocco; Amal.amaghnouje@usmba.ac.ma (A.A.); Imane.essafi1@usmba.ac.ma (I.E.-S.); dalila.bousta@usmba.ac.ma (D.B.); 2Institute of Organic Chemistry, National Academy of Sciences, 02660 Kyiv, Ukraine; bogza@i.ua (S.B.); nat2bogdan@gmail.com (N.B.); iamkaant@gmail.com (A.K.); 3Laboratory of Toxicology, Laboratory of Medical Analysis, Hassan II University Hospital, Fez 30050, Morocco; sanae.achour@usmba.ac.ma; 4Department of Pathology, Hassan II University Hospital, Fez 30050, Morocco; hinde.elfatemi@usmba.ac.ma; 5Department of Chemistry, Materials Science Division, University of Helsinki, 00560 Helsinki, Finland

**Keywords:** benzodiazepines, central nervous system, mouse behavioral model, toxicity

## Abstract

We report the design and synthesis of a new diazepine derivative, 4-(4-methoxyphenyl)-2,3,4,5-tetrahydro-2,3-benzodiazepin-1-one (VBZ102), and the evaluation of its anxiolytic-like profile, memory impairment effect, and toxicity in Swiss mice. VBZ102 was evaluated for central nervous system effects in an open field, light–dark box, and novel object recognition tests under oral administration for acute and sub-acute treatment. We tested the VBZ102 toxicity in mice through a determination of LD_50_ values and examination of the biochemical and histopathological parameters. The VBZ102 induced an anxiolytic effect at different doses both in the light–dark box and open field tests. Unlike other benzodiazepines (e.g., bromazepam), a sedative effect was noted only after administration of the VBZ102 at 10.0 mg/kg.

## 1. Introduction

Anxiety is a normal physiological state that is beneficial in certain dangerous life situations and triggers the fight or flight stress response and physical symptoms resulting from the autonomic nervous system response. However, anxiety may transform into a mental illness when it becomes chronic, irrational, and interferes with many life functions [1]. A majority of drugs used in anxiety treatment represent positive modulators of the gamma-aminobutyric acid (GABA) transmission, including benzodiazepine derivatives, and selective serotonin reuptake inhibitors (SSRIs) of different chemical natures [2,3].

The interest of medicinal chemists in 2,3-benzodiazepines (2,3BDZs) has exhibited an exponential growth since the discovery of their action as noncompetitive α-amino-3-hydroxy-5-methyl-4-isoxazolepropionic acid (AMPA) receptor antagonists endowed with anticonvulsant and neuroprotective properties. A number of active molecules of this type have been synthesized over the past 40 years. Some of those compounds, e.g., tofisopam (Grandaxin^®^), girisopam, nerisopam, possess significant anxiolytic and antipsychotic activities (Figure 1) [4]. Along with those, the compounds in question are known to have unwanted side-effects, such as an anterograde memory loss [5,6]. Therefore, there is a pressing need for the development of new active compounds with favorable side-effect profiles, credible benefits, and moderate costs. In the search for alternative and more specific molecules, current investigations are focused on the anxiolytic, sedative, and locomotor activities of various benzodiazepine derivatives. For example, new potent anxiolytic dibenz-(1,4)-diazepin-1-one derivatives were recently synthesized [7].

The biological activity of a molecule is largely governed by its geometry. The latter itself is mainly controlled by a core scaffold, and the substituents play only a minor role [8,9]. However, current commercial databases used for drug discovery lack 3D scaffolds and mostly consist of linear or planar sp^2^-rich ones [9,10]. Therefore, the development of new geometrically diverse scaffolds is of vital importance. 2,3-Benzodiazepines have a non-planar structure and show promising results, particularly for their 3,4-dihydro-derivatives [11]. In particular, a drug candidate talampanel have shown prominent anticonvulsant activity [12].

All known successes in the development of noncompetitive AMPA receptor antagonists were based on modifications of the 1-aryl-2,3-benzodiazepine scaffold. Recently, we published a method of synthesis of 4-aryl-2,3-benzodiazepin-1-ones, which are the regioisomers of 1-aryl-2,3-benzodiazepine-4-ones [13].

The present paper is devoted to a new central nervous system (CNS)-active benzodiazepine derivative. We report the synthesis, pharmacological activity in mouse behavioral models, and toxicity evaluation of the new compound 4-(4-methoxyphenyl)-2,3,4,5-tetrahydro-2,3-benzodiazepin-1-one (VBZ102).

## 2. Results

### 2.1. Chemical Synthesis

2,3-Benzodiazepin-1-ones **1a–c** (Figure 2) were prepared according to our previously reported procedure [14]. Literature data on the reduction of 2,3-benzodiazepin-1-ones are not available. Grasso et al. [15] described the reduction of 2,3-benzodiazepin-4-ones using sodium cyanoborohydride in methanol. We investigated the reduction of 4-(4-methoxyphenyl)-2,5-dihydro-1*H*-2,3-benzodiazepin-1-one **1b** with cyanoborohydride, borohydride, and sodium trimethoxyborohydride. It was found that the reduction of the C(4)=N(3) double bond of the diazepine **1b** proceeded selectively, and the yield of the diazepine **2b** reached 40%. The reaction of **1b** with tris-(methoxy)sodium borohydride led to a complex mixture of products, while the use of sodium borohydride gave the starting compound only.

We developed a simple and effective method of C=N bond reduction in benzodiazepines **1a–c** by Pd-catalyzed hydrogenation. The reaction was carried out in methanol at room temperature, an aqueous solution of hydrochloric acid was added to improve the solubility. The yields of diazepines **2a–c** reached 80–90%. Both analytical and spectral data of all synthesized compounds were in full agreement with the proposed structures; stereoselectivity of the reaction was not studied.

Taking into consideration the existing pharmacological data for the compounds of type **1** [16], we selected 4-(4-methoxyphenyl)-2,3,4,5-tetrahydro-2,3-benzodiazepin-1-one **2b** (VBZ102) for activity and toxicity evaluation. On one hand, that choice was also supported by the presence of only one substituent, *viz.* the methoxy group, which had been shown to decrease the toxicity of drugs [17,18]. On the other hand, the VBZ102 allowed us to test the activity of the scaffold itself that was not biased by the substitution pattern.

### 2.2. VBZ102 Effect in the Experimental Anxiety Tests

#### 2.2.1. Acute Treatment

To determine the effective dose, we used three doses of the VBZ102 in mice P.O. (0.5, 1.0, and 10.0 mg/kg bw) in the open field and LDB tests. The results are compared with vehicle, negative control, and bromazepam, a reference drug.

##### Open Field Test

We studied the effect of increasing doses of the VBZ102 on the number of total squares crossed and the time spent in the center of the open field test box.

The amount of 10.0 mg/kg of orally administered VBZ102 showed a significant diminution of the total squares crossed (F = 169.8; *p* < 0.001). However, no considerable difference was observed for the doses of 0.5 mg/kg and 1.0 mg/kg of the VBZ102 when compared to the vehicle (Figure 3A).

The time spent in the center was increased at the doses of 10.0 and 1.0 mg/kg of the VBZ102 when compared to the vehicle and bromazepam (F = 15.82; *p* < 0.001) (Figure 3B).

##### Light–Dark Box Test

VBZ102 (1.0 and 10.0 mg/kg) and bromazepam (1.0 mg/kg) led to a significant increase of the time spent in the light compartment (F = 433.8; *p* < 0.001), when compared to the vehicle (Figure 3C). Extended times spent in the light compartment were also observed for the administration of 0.5 mg/kg of the VBZ102.

However, the number of transitions between the light and dark compartments was also increased (F=30.95; *p* < 0.001) under treatment with 1.0 mg/kg of the VBZ102, followed by bromazepam (1.0 mg/kg) and 0.5 mg/kg dose of the VBZ102 (Figure 3D).

#### 2.2.2. Chronic Treatment

##### Open Field Test

In the open field test, administration of 1.0 mg/kg of the VBZ102 for 1, 7, 14, and 21 days significantly increased the time spent in the center area (Figure 4A) but did not affect the number of total squares crossed (F (6,48) = 4.335; *p* = 0.001) (Figure 4B).

The VBZ102 also substantially increased the time spent in the center area with respect to the same dose of bromazepam (Figure 4A), but there was little difference in the total squares crossed (F (6,24) = 13.14; *p* < 0.001) (Figure 4B).

##### Light–Dark Box Test

Treatment with 1.0 mg/kg of the VBZ102 increased the number of transitions and the time spent in the light area when compared to the negative control. The time spent in the light area was almost identical during the treatment period. However, the number of transition was higher in the 21 days term (F (6, 48) = 1.992; *p* = 0.085; F (6, 48) = 1.53; *p* = 0.188) (Figure 4C,D).

There was no significant difference of the time spent in the light area in comparison to bromazepam. However, the number of transitions between the light and dark compartments was increased for the mice treated with the VBZ102.

### 2.3. Evolution of VBZ102 Side Effects on Memory

#### Novel Object Recognition

During the training period, all groups spent a similar time exploring the two different objects. In addition, there were no significant changes in the total exploration time. During the testing period the administration of bromazepam decreased the exploration time. However, a comparison of the exploration time of the familiar object vs. the novel one did not show any substantial difference among the groups (Figure 5). Administration of bromazepam decreased the time spent to explore a new object (F (6, 48) = 11.8; *p* < 0.001). Furthermore, we noted that the discrimination index for the bromazepam group was clearly lower than that of the VBZ102 group (F (6, 48) = 60.93; *p* < 0.001).

### 2.4. Toxicity

In order to assess chronic toxicity, the VBZ102 was administered orally at the doses of 10.0 mg/kg and 1.0 mg/kg for 21 days. No mortality and no poisoning symptoms were observed, except for a mildly reduced mobility reaction for the mice treated at the dose of 10.0 mg/kg. Iridescent hair was observed during the first week of the treatment, but it disappeared subsequently.

Concurrently, after 28 days, no noticeable body weight difference was observed for the mice of the control group and those treated with 10.0 and 1.0 mg/kg doses (Table 1). The relative weight of the spleen was significantly (*p* < 0.001) increased in the group treated with 10.0 mg/kg of the VBZ102 (F (2.6) = 89,34; *p* < 0.001), whereas that of the other organs remained unchanged (Table 2). Thus, our data indicate that VBZ102 did not cause any adverse effects on the body weight.

#### Biochemical Parameters

Biochemical tests showed no significant differences, except for the aspartate aminotransferase (AST) test (F (2.6) = 0.875 *p* = 0.650, F (2.6) = 0.463 *p* = 0.649, F (2.6) = 2.723 *p* = 0.144) for the urea, creatinine, and ALT, respectively. The AST values increased significantly in the group treated with the 10 mg/kg dose when compared to the vehicle (F (2.6) = 95.22 *p* < 0.001) (Table 3).

### 2.5. Histological Examination

Microscopic examination of sections of different organs of the control and treated groups showed an absence of any gross pathological lesions (Figure 6) and a normal liver histology at the dose of 1.0 mg/kg. However, the presence of nuclear degenerescence, vacuolation, and steatosis was noticed in the liver cells at the dose of 10 mg/kg, and the nuclei were not centered inside the cells.

## 3. Discussion

The aim of this study was to evaluate the VBZ102 effect on anxiety in mice. The doses of 0.5 mg/kg, 1.0 mg/kg, and 10.0 mg/kg were chosen to select an effective dose that would not trigger side-effects. The dose of 0.5 mg/kg produced no significant effect. Administration of 1.0 mg/kg of VBZ102 led to an increase both in the number of entries and the time spent in the light area in the LDB test, and in the time spent in the center and number of total squares crossed in the open field. While the time spent in the light area in the LDB test and that in the center in the open field test increased after the dose of 10.0 mg/kg, this dose, however, provoked a decrease in both the number of transitions in the LDB test and the number of total squares crossed in the open field test, indicating a sedative effect. Therefore, we chose 1.0 mg/kg of VBZ102 as the effective dose and continued the sub-acute study at that dose in the LDB and open field tests. The higher dose of 10.0 mg/kg was used to investigate possible undesirable effects, such as memory alteration and toxicity.

The open field test is a very popular animal model of anxiety-like behavior. It is used to assess the locomotor behavior of mice and to detect anxiogenic- or anxiolytic-like agents [19]. The results showed that the VBZ102 increased the time spent by the mice in the central area of the open field test in a dose-dependent manner. At 1.0 mg/kg, the compound under investigation induced an increase both in time spent in the central area and the total number of square crossings, which suggests an anxiolytic-like activity. The treatment (1.0 mg/kg) did not change the total number of crossings, the immobility time, or the number of rearing behaviors, thereby suggesting no alteration in the locomotion of animals.

In the light–dark box test, anxiety is generated by the conflict between desire to explore and to retreat from an unknown and well-illuminated space [20]. Our experimental data showed an increase in both the time spent in the light compartment and in the number of transitions between the LDB chambers after a single dose of the VBZ102. Moreover, after treatment for 21 days, the number of transitions between the compartments and the time spent in the light one significantly increased. Therefore, the VBZ102 revealed a pronounced anxiolytic effect like that of bromazepam. The anxiolytic activity mechanism of both compounds could either be related to GABA A receptor complex or may be similar to 2,3-benzodiazepine AMPA antagonists [11]. Our results corroborate those obtained for other 2,3-benzodiazepine AMPA receptor antagonists (e.g., GYKI 53405 [21], GYKI 53655 [22], and EGIS-8332 [23,24]) that demonstrated their anxiolytic effects in several behavioral tests, such as the LDB, elevated plus maze (EPM), and Vogel tests [25]. The toxicity study revealed a significant increase in the values of AST and some change in the histological sections of liver at the dose of 10.0 mg/kg (see Figure 6B). Therefore, the VBZ102 at 10.0 mg/kg is capable of inducing lesions in the liver. The relative organ weight study showed a significant increase of spleen in mice, which can be explained by leukocytosis, extramedullary hematopoiesis, erythrophagocytosis, or septic shock [21]. However, no significant changes were observed in biochemical and histological parameters at the dose of 1.0 mg/kg.

It is interesting to note that VBZ102 had no sedative effect at the dose of 1.0 mg/kg, while the bromazepam is known as an anxiolytic and sedative drug [26,27]. This observation was confirmed by the increase in the number of transitions in the LDB test and no change in the number of total squares crossed in the open field one.

To the best of our knowledge, undesirable side-effects of diazepines on memory have not been studied, despite some negative effects being demonstrated [6,28]. In the present study, the VBZ102 effects on learning and memory impairments were investigated. The memory function of mice treated with VBZ102 for 21 days remained intact when compared to bromazepam, which provokes a decrease in memory and learning capacity in the novel object recognition test. Our data clearly demonstrated that the compound under investigation did not show side-effects on memory in the novel object recognition tests.

## 4. Materials and Methods

### 4.1. Chemical Synthesis

#### 4.1.1. General Experimental Procedures

The reagents and solvents were obtained from commercial suppliers and used without further purification unless otherwise specified. All reagents were weighed and handled in air at room temperature, and the reactions were performed in round bottomed flasks. Organic solutions were concentrated on a rotary evaporator at 23–35 °C. Melting points (uncorrected) were determined with a capillary melting point apparatus. Proton and carbon nuclear magnetic resonance (^1^H and ^13^C NMR) spectra were recorded with a Varian VXR-300 (299.9 MHz) spectrometer in parts per million from internal tetramethylsilane on the δ scale and were referenced from the residual proton or carbon resonances in the NMR solvent (DMSO: ^1^H δ 2.50, ^13^C δ 39.5). The spectral data are reported as follows: chemical shift, multiplicity s = singlet, d = doublet, t = triplet, m = multiplet.

#### 4.1.2. 4-Phenyl-2,3,4,5-tetrahydro-1*H*-2,3-benzodiazepin-1-one (2a)

M.p. 132–135°C. Yield 80%. ^1^H NMR (DMSO-d_6_, 300 MHz): 2.89–2.94 (1H, m, CH); 3.18–3.22 (1H, m, CH); 4.22–4.23 (1H, m, CH); 5.46 (1H, m, NH); 7.23–7.49 (8H, m, CH); 7.64–7.69 (1H, m, CH); 9.15 (1H, s, NH). ^13^C NMR (DMSO-d_6_, 75 MHz): 30.11; 63.85; 126.60; 126.64; 126.78; 128.02; 128.80; 130.66; 135.31; 136.74; 143.22; 174.46 (C=O).

#### 4.1.3. 4-(4’-Methoxyphenyl)-2,3,4,5-tetrahydro-1*H*-2,3-benzodiazepin-1-one (2b)

M.p. 135–138°C. Yield 80%. ^1^H NMR (DMSO-d_6_, 300 MHz): 2.86–2.92 (1H, m, CH); 3.18–3.23 (1H, m, CH); 3.75 (3H, s, OCH3); 4.21–4.27 (1H, m, CH); 5.37 (1H, m, NH); 6.89 (2H, d, J = 8Hz); 7.30 (1H, d, J = 8); 7.35 (2H, d, J = 8); 7.40(1H, t, J = 8); 7.47 (1H, t, J = 8); 7.65 (1H, t, J = 8); 9.12 (1H, s, NH). ^13^C NMR (DMSO-d_6_, 75 MHz): 30.11; 63.85; 126.60; 126.64; 126.78; 128.02; 128.80; 130.66; 135.31; 136.74; 143.22; 174.46 (C=O).

#### 4.1.4. 4-(2,5-Dimethoxyphenyl)-2,3,4,5-tetrahydro-1*H*-2,3-benzodiazepin-1-one (2c)

M.p. 105–107°C. Yield 85%. ^1^H NMR (DMSO-d_6_, 300 MHz): 2.75–2.80 (1H, m, CH); 3.08–3.12 (1H, m, CH_2_); 3.75 (3H, s, OCH_3_); 3.80 (3H, s, OCH_3_); 4.34–4.39 (1H, m, CH); 5.37 (1H, br.s, NH); 6.44–6.47 (1H, m, CH); 6.54–6.55 (1H, m, CH); 7.20–7.22 (1H, m, CH); 7.20–7.47 (3H, m, CH); 7.63–7.64 (1H, m, CH); 9.15 (1H, s, NH). ^13^C NMR (DMSO-d_6_, 75 MHz): 55.10 (CH_3_); 55.45 (CH_3_); 57.91 (CH_2_); 90.86; 104.28; 126.60; 127.43; 128.62; 128.68; 130.62; 135.29; 136.92; 156.52; 159.22; 174.37 (C=O).

### 4.2. Animals

Male Swiss mice (weighing 25–35 g) from the Pasteur Institute (Rabat, Morocco), were used for the tests. The animals were caged in groups of 20 mice/cage, at room temperature of 23 ± 2 °C, with free access to tap water and food under a 12:12 h light/dark cycle (lights on at 06:00 a.m.). All manipulations were carried out between 8:00 a.m. and 3:00 p.m., and each animal was used only once.

All the experimental procedures were carried out in compliance with the biosecurity clauses stipulated by the ethical guidelines on animal experimentation to guarantee the use of the laboratory animals under optimal conditions and, at the same time, to ensure the animal’s safety. All surgical equipment and operating room storage devices were autoclaved before each use.

The concept of 3R (Reduce, Refine, Replace) established by Russel and Burch in 1959 [29] was taken into account when planning the experiments. With respect to the principle, we used a smaller number of animals in each experimental group. Moreover, euthanasia was carried out under anesthesia to minimize the animal’s suffering.

### 4.3. Anxiety: Acute and Subacute Treatment

#### 4.3.1. Acute Treatment

The unique oral administration of the vehicle (2% Tween 20, NaCl, 10.0 mL/kg), bromazepam (the reference drug, 1.0 mg/kg), and VBZ102 (0.5, 1.0 and 10.0 mg/kg) was carried out for three groups of five animals.

#### 4.3.2. Subacute Treatment

The oral administration of the vehicle (2% Tween 20, NaCl, 10.0 mL/kg), bromazepam (1.0 mg/kg), and VBZ102 (1.0 and 10.0 mg/kg) was carried out for three groups of five animals once a day for 21 days

### 4.4. Behavioral Tests

#### 4.4.1. Open Field Test

The test was used to evaluate anxiety, exploration, and locomotion. After 60 min of the drug or vehicle administration, the test was performed by placing each mouse in a center square to explore the arena and the numbers of ambulation, rearing, and crossing of the central squares were recorded using a digital video camera [30,31].

Central squares are those that are not adjacent to the arena walls [32]. The square number was counted when the mouse entered a square with all four paws [33]. Ambulation means the total number of squares crossed by the mouse, and the crossing of the central square is the number of times the mouse entered the central squares with all its four paws [34].

#### 4.4.2. Light–Dark Box Test (LDB)

The light–dark box was a wooden box with dimensions of 44 × 21 × 21 cm and was divided into two compartments: the first one was painted black inside and covered with a wooden lid, and the second one was painted white. The two compartments were separated by a wooden blank, having a hole of 7 × 7 cm in the center at the floor level.

After 60 min of the drug or vehicle administration, a 5 min test was performed by placing each mouse in the center of the dark box [35], keeping its face away from the opening hole. The number of transitions and time spent in each compartment were recorded using a digital video camera [36].

#### 4.4.3. Novel Object Recognition Test

The distinction between familiar and unfamiliar objects is an index of recognition memory. The measurement is based on an innate preference of rodents for novel objects over familiar ones [37,38].

The mice were divided into three groups of five animals and received an oral dose of the vehicle (2% Tween 20, NaCl, 10.0 mL/kg), bromazepam (1.0 mg/kg), or VBZ102 (10.0 mg/kg) treatment on day 21. After 60 min of the drug or vehicle administration, the training trial was performed. The mice were placed in a 52 × 52 × 25 cm box and allowed to explore two identical objects for 5 min each. The training was repeated twice. The test trial was performed 24 h after the training. Then, one of the two familiar objects (F) was replaced with a new one (N), and the mice were allowed to explore them for 5 min. Discrimination index D was used as a measure of the object recognition [39]:(1)$$DI=\frac{Tn-Tf}{Tn+Tf}$$
where *Tn* is the exploration time devoted to the novel object, and *Tf* is the exploration time devoted to the familiar object.

### 4.5. Toxicity

Body weight and local injuries were studied throughout the treatment. Mortality, if any, was recorded in all groups during the treatment. At the end of the treatment (28 days), biochemical (liver and renal function assays) and histological parameters were investigated. The organs were quickly blotted, weighed on a digital balance, and processed for histological studies. The organ/body weight ratio was calculated for each organ, and tissues were processed by H&E staining [40].

#### 4.5.1. Biochemical Parameters

Biochemical assays (urea, creatinine, alanine aminotransferase (ALT), aspartate aminotransferase (AST)) were performed using the blood plasma and Randox kits at the end of the assay [41,42].

#### 4.5.2. Histological Examination

At the end of the treatment, the animals were euthanized, and several organs, e.g., liver, kidneys, and brain were collected for histological examinations. All the organs were immediately fixed in 10% buffered formalin and processed with H&E staining for histology [43,44].

### 4.6. Statistical Analyses

Statistical analysis was performed in GraphPad Prism software using a one-way ANOVA followed by Dunnett’s post hoc test to describe the toxicity results and a two-way ANOVA followed by Tukey’s post hoc test to describe the results of anxiety and memory tests (*p* < 0.05).

## 5. Conclusions

We propose a new synthetic approach to 2,5-dihydro-1*H*-2,3-benzodiazepin-1-ones. One representative of the newly synthesized diazepines, viz. the 4-(4-methoxyphenyl)-2,3,4,5-tetrahydro-2,3-benzodiazepin-1-one (VBZ102), was tested for CNS activity and side-effects in mice. It was shown that it has a pronounced anxiolytic-like effect in different model experiments without provoking memory impairments and compromising the motor activity and biochemical parameters of the experimental animals. Unlike other benzodiazepines (e.g., bromazepam), a sedative effect was noted only after administration of the VBZ102 at 10.0 mg/kg.

## Figures and Tables

**Figure 1 pharmaceuticals-14-00814-f001:**
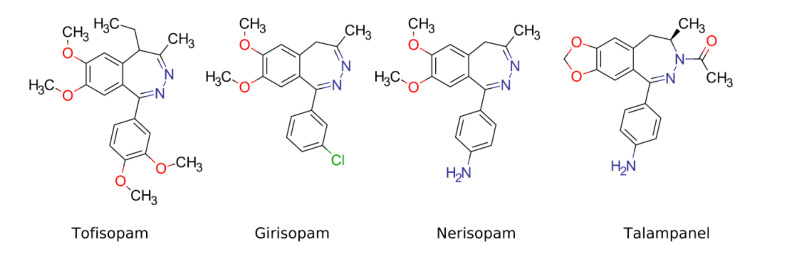
2,3-Benzodiazepines with anxiolytic activity.

**Figure 2 pharmaceuticals-14-00814-f002:**
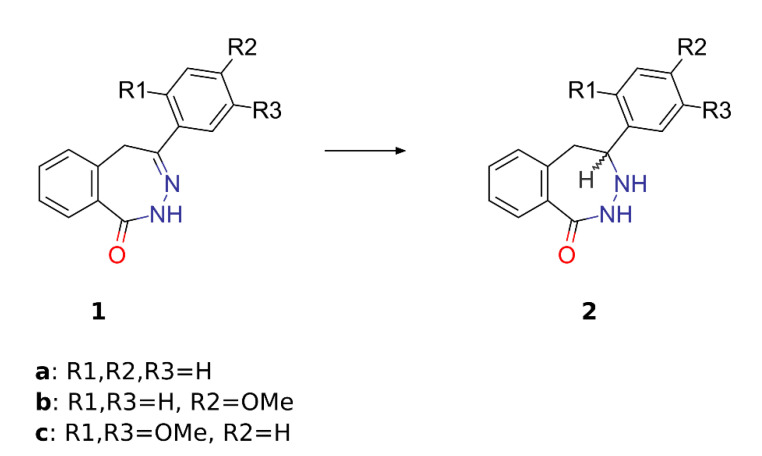
Hydrogenation of 2,3-benzodiazepin-1-ones. Reagents and conditions: H_2_, 10 % Pd/C, MeOH-HCl, r.t., atmospheric pressure.

**Figure 3 pharmaceuticals-14-00814-f003:**
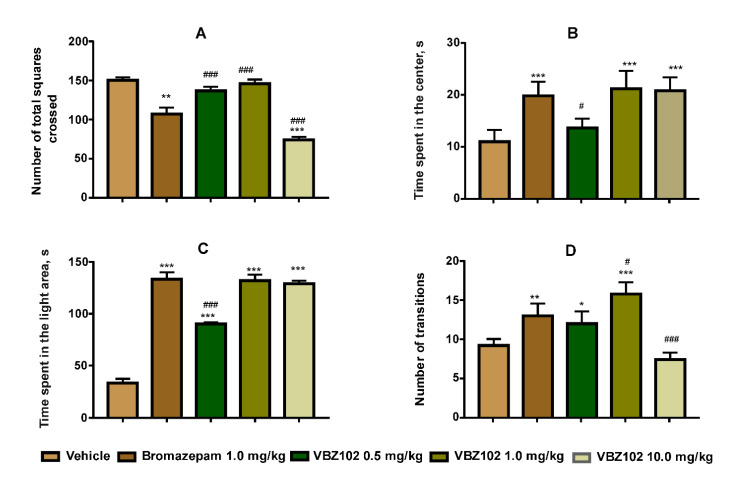
Effects of VBZ102, vehicle, and bromazepam on (**A**) the number of total squares crossed in the open field test, (**B**) the time spent in the center in the open field test, (**C**) the time spent in the light area in the LDB test, and (**D**) the number of transitions in the LDB test. The data are expressed as mean ± standard deviation. The analysis was done using two-way repeated-measures analysis of variance followed by the Tukey post-hoc test. *** *p* ≤ 0.001, ** *p* ≤ 0.01, * *p* ≤ 0.05, compared to the respective session of the vehicle group. ^###^ *p* ≤ 0.001, ^#^ *p* ≤ 0.05, compared to the bromazepam group.

**Figure 4 pharmaceuticals-14-00814-f004:**
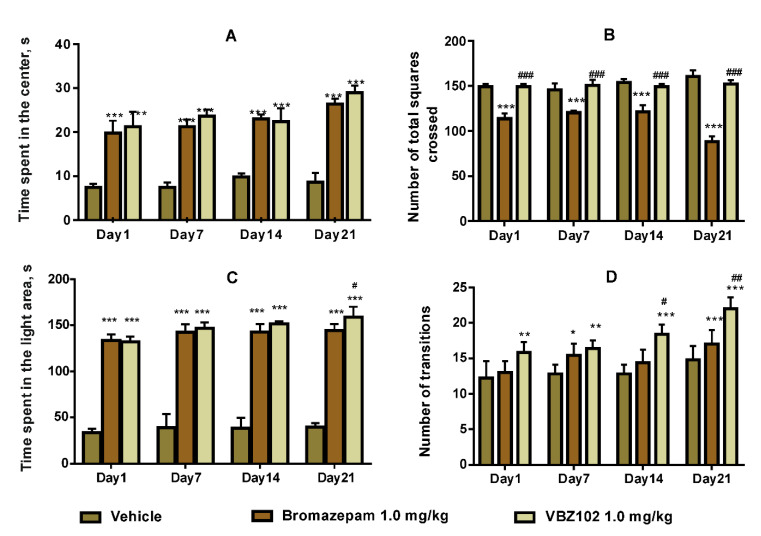
Effects of the VBZ102, vehicle, and bromazepam on (**A**) the time spent in the center in the open field test, (**B**) the number of total squares crossed in the open field test, (**C**) the time spent in the light area in the LDB test, and (**D**) the number of transitions between the light and dark compartments in the LDB test. The data are expressed as mean ± standard deviation. The analysis was done using two-way repeated-measures analysis of variance followed by the Tukey post-hoc test. *** *p* ≤ 0.001, ** *p* ≤ 0.01, * *p* ≤ 0.05, compared to the respective session of the vehicle group. ^###^
*p* ≤ 0.001, ^##^ *p* ≤ 0.01, ^#^ *p* ≤ 0.05, compared to the bromazepam group.

**Figure 5 pharmaceuticals-14-00814-f005:**
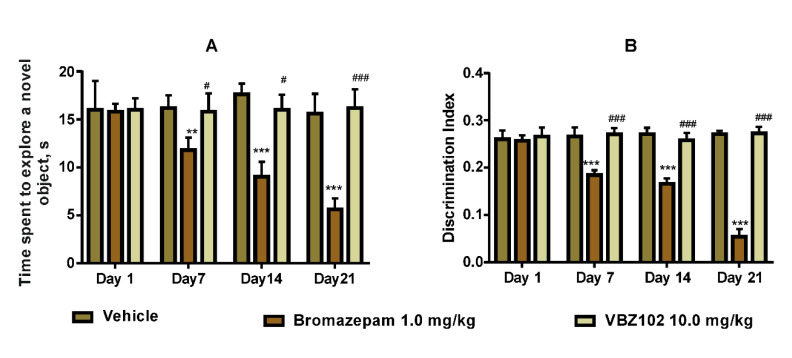
Novel object recognition test. (**A**) Time spent by different groups of mice to explore the novel object. (**B**) Discrimination index performance of different groups of mice. The data are expressed as mean ± standard deviation. The analysis was done using two-way repeated-measures analysis of variance followed by the Tukey post-hoc test. *** *p* ≤ 0.001, ** *p* ≤ 0.01, compared to the vehicle group, ^###^
*p* ≤ 0.001, ^#^ *p* ≤ 0.05, compared to the bromazepam group.

**Figure 6 pharmaceuticals-14-00814-f006:**
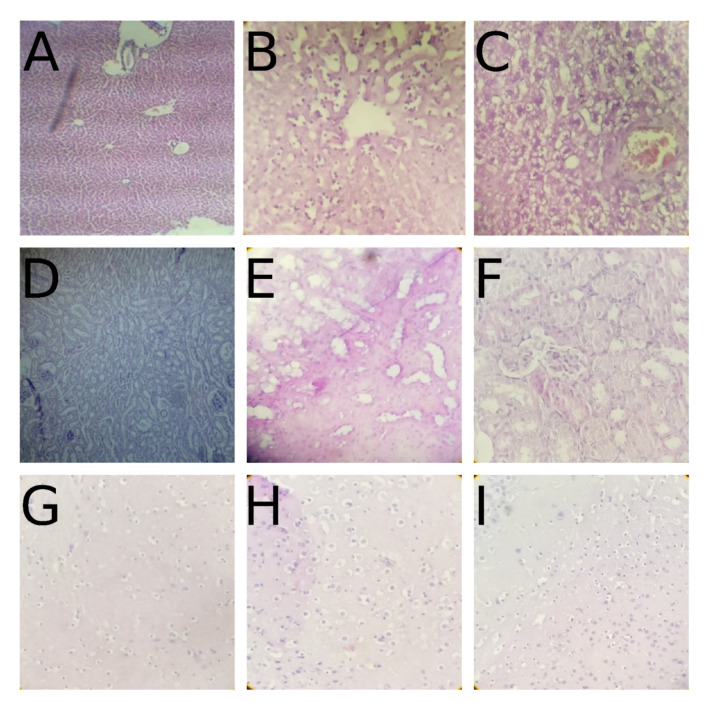
Histopathological examination of liver, kidneys, and brain of mice in the chronic oral toxicity study. (**A**,**D**,**G**) The control group; (**B**,**E**,**H**) the group treated with 10.0 mg/kg of the VBZ102; (**C**,**F**,**I**) the group treated with 1.0 mg/kg of the VBZ102. (**A**,**D**,**G**) Normal cells. (**B**) Nuclear degenerescence, vacuolation, and steatosis within cells; the nuclei are not centered, although, there is no inflammatory infiltrate around the centrilobular vein. (**E**) Normal cells. (**H**) Presence of a clear halo around perivascular halo vessels. (**C**) Degeneration of the plasma membrane, vacuolation, steatosis, and infiltrate around the centrilobular vein. (**F**) Normal cells. (**I**) Normal cells (hematoxylin and eosin (H&E) ×400).

**Table 1 pharmaceuticals-14-00814-t001:** Effects of the orally administered VBZ102 on body weight (g) after 28 days of treatment.

	Week 1	Week 2	Week 3	Week 4
Vehicle	35.2 ± 1.2	36.0 ± 1.2	38.0 ± 2.2	37.4 ± 1.4
VBZ102, 1.0 mg/kg	31.2 ± 2.4	32.4 ± 1.3	32.2 ± 3.1	33.2 ± 1.5
VBZ102, 10.0 mg/kg	33.4 ± 2.2	32.2 ± 4.0	30.4 ± 2.3	37.0 ± 1.2

Note: Values are expressed as mean ± SD of 5 mice in each group. One-way ANOVA followed by Dunnett’s post hoc test are used; *p* < 0.001 compared to control.

**Table 2 pharmaceuticals-14-00814-t002:** Effect of the orally administered VBZ102 on average relative organ weight (%) at the 28th day of treatment.

	Liver	Kidneys	Spleen	Adrenal Glands	Lungs
Vehicle	10.10 ± 0.67	1.62 ± 0.07	0.77 ± 0.09	0.40 ± 0.04	0.51 ± 0.10
VBZ102, 1.0 mg/kg	9.71 ± 0.49	1.92 ± 0.03	0.80 ± 0.12	0.03 ± 0.03	0.56 ± 0.02
VBZ102, 10.0mg/kg	9.13 ± 0.79	1.31 ± 0.24	5.65 ± 0.24 ^a^	0.03 ± 0.00	0.63 ± 0.08

Note: Values are expressed as mean ± SD of 5 mice in each group. One-way ANOVA followed by Dunnett’s post hoc test are used. ^a^
*p* < 0.001 compared to control.

**Table 3 pharmaceuticals-14-00814-t003:** Effect of the orally administered VBZ102 on the biochemical parameters of mice.

	Vehicle	VBZ102, 1.0 mg/kg	VBZ102, 10.0 mg/kg
Urea	0.28 ± 0.02	0.24 ± 0.01	0.38 ± 0.04
Creatinine	3.40 ± 0.31	4.00 ± 0.00	4.33 ± 0.33
ALT	45.80 ± 1.11	23.00 ± 2.52	44.33 ± 12.72
AST	311.00 ± 27.22	223.33 ± 13.33	568.00 ± 180.58 ^a^

Note: Values are expressed as mean ± SD of 5 mice in each group. One-way ANOVA followed by Dunnett’s post hoc test are used; ^a^
*p* < 0.001 compared to control.

## Data Availability

The data supporting the conclusions of this article are included within the article.

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
