# Peer review of "New 2,3-Benzodiazepine Derivative: Synthesis, Activity on Central Nervous System, and Toxicity Study in Mice"

_pharmaceuticals, 2021, doi:10.3390/ph14080814_

Round 1

Reviewer 1 Report

The paper of Amaghnouje et al describes the design, synthesis and biological evaluation of three new benzodiazepine derivatives. The work is interesting enough, but it has several issues to clarify.

Typographical errors:

Line 39: not "nerisopamexert possess" but "nerisopam possess"

Line 70: at 23… .35 ° C. What does it mean? If it is a range, mark it as 23 - 35 ° C.

Line 182: at 80… 90%. Like above. What does it mean?

In figure 3: there is no correspondence between the legend of figures A and B and the related figures which are inverted.

Major issues

- There is confusion in the NMR data. First of all, the 1H-NMR data entered in lines 78-80 do not correspond to compound 2a, which does not have the OCH3 group. So the description of compound 2b is also wrong.

The NMR data includes two CH groups and one CH2 group (there is only one CH group and one CH2 group). Only one NH group is reported.

All NMR data should be carefully checked.

- Was the discrimination index formula obtained from the literature (then the reference is lacking) or was it set by the Authors?

- A reference drug is missing in acute treatment

- In the number of transitions in the LDB test the dosage of 10 mg/kg drastically decreases the number. The Authors did not provide an interpretation of the data.

- In open field test: line 224-225: the administration of MPDT1 has increased the number of total squares crossed when compared to the vehicle. In reality the graphic shows an almost constant value (just above 150).

- Table 2: the dose of 10 mg/kg of MPTD1 significantly increases the weight of the spleen. Authors should explain what kind of effect can justify this effect, since they claim poor toxicity.

- Line 318: it is interesting to note that the MPTD1 had no sedative effect. Authors should better justify and explain this claim.

The major issue to be clarified concerns the stereochemistry of the compounds obtained. In fact, following the reduction, a chiral carbon is formed. Authors should clearly enter whether the stereo selectivity of the reaction has been studied. As talampanel and many other CNS active drugs demonstrate, often only one enantiomer is highly active and has optimal interaction with biological targets (receptors). In other words, the Authors cannot overlook and fail to mention this problem.

Author Response

please, see attachment

Reviewer 2 Report

               I have read the manuscript entitled “New Benzodiazepine Derivative: Synthesis, Activity on Central Nervous System, and Toxicity Study in Mice” (ID: pharmaceuticals-1276608) and I think that it is a valuable and interesting paper that present the influence of a new diazepine derivative 4-(4-methoxyphenyl)-2,3,4,5-tetrahydro-2,3-benzodiazepin-1-one (MPTD1) on anxiety-like behaviour and memory in mice. The Authors evaluated also some possible toxic effects of this compound in mice.                In my opinion, the topic raised in the manuscript is interested to but the manuscript must be significantly improved. In the current version of the manuscript, there are many mistakes related to the description of research methods and obtained results, as well as the analysis and interpretation of these results. The manuscript does not meet the requirements of Pharmaceuticals and cannot be considered further for publication in this journal. My main comments regarding this manuscript are included below:1.      First of all, I have serious concerns about whether groups of 5 animals are large enough to produce reliable behavioural test results. In my opinion, the 3R rule should not be given priority over the reliability and validity of the results obtained.2.      Animal treatment with the studied compounds should be described precisely (i.e., both acute and chronic) in a separate subchapter in the “Materials and Methods” section. In the present version of the manuscript Authors did not mention the acute treatment of animals, there is the only description of the chronic treatment. Moreover, the description of animal treatment should be removed from subsections that describe behavioural tests. 3.      Statistical analysis conducted in the study should be described in more detail, i.e., which results were analyzed using one-way ANOVA and which ones with two-way ANOVA. Moreover, in “Results” subsection, the Authors compare some results from the MPTD1-treated group with results from the bromazepam-treated group (i.e., line 226-228: “The MPDT1 has also substantially increased the time spent in the center area with respect to the same dose of bromazepam (Fig. 4A), but there was a little difference in the total squares crossed”; line 240-241: “There was no significant difference of the time spent in the light area in comparison to bromazepam”) while Dunnett’s test was used for statistical analysis. This pos hoc test allows comparison of the studied groups (bromazepam- and MPTD1-treated) only with the control (i.e., vehicle-treated) group.  4.      Description of the results should be significantly improved. There is a lots of mistakes and imprecise terms. For example:·        line 224-225: “….and the number of total squares crossed (Fig. 4B), when compared to the vehicle” – according to the data presented in Figure 4B, number of total squares crossed was not affected by MPTD1 treatment. ·        Line 246-248: “During the testing period, both the vehicle and the MPTD1-treated mice showed significant increase in the exploration time” - rather, the administration of bromazepam decreased the exploration time. ·        Line 260-261: “…except for a mildly reduced mobility reaction for the mice treated at the dose of 260 10.0 mg/kg” – how the mobility was evaluated?·        Subsection 3.5. – the Authors mentioned that MPTD1 at a dose of 1 mg/kg did not provoke any pathological lesions. What about a dose of 10 mg/kg??????5.      Authors did not provide the results of one-way and two-way ANOVA. The F and p values should be included in the “Results” part.6.      The discussion should be completed. Authors did not refer at all to the results of experiments evaluating the toxicity of MPTD. The results obtained in behavioural tests were not compared with the effects of other known non-competitive AMPA receptor antagonists.  

Author Response

Reviewer 2, round 1

Open Review

English language and style

( ) Extensive editing of English language and style required
(x) Moderate English changes required
( ) English language and style are fine/minor spell check required
( ) I don't feel qualified to judge about the English language and style

Yes

Can be improved

Must be improved

Not applicable

Does the introduction provide sufficient background and include all relevant references?

( )

(x)

( )

( )

Is the research design appropriate?

( )

( )

(x)

( )

Are the methods adequately described?

( )

( )

(x)

( )

Are the results clearly presented?

( )

( )

(x)

( )

Are the conclusions supported by the results?

( )

( )

(x)

( )

Comments and Suggestions for Authors

I have read the manuscript entitled “New Benzodiazepine Derivative: Synthesis, Activity on Central Nervous System, and Toxicity Study in Mice” (ID: pharmaceuticals-1276608) and I think that it is a valuable and interesting paper that present the influence of a new diazepine derivative 4-(4-methoxyphenyl)-2,3,4,5-tetrahydro-2,3-benzodiazepin-1-one (MPTD1) on anxiety-like behaviour and memory in mice. The Authors evaluated also some possible toxic effects of this compound in mice.

In my opinion, the topic raised in the manuscript is interested to but the manuscript must be significantly improved. In the current version of the manuscript, there are many mistakes related to the description of research methods and obtained results, as well as the analysis and interpretation of these results. The manuscript does not meet the requirements of Pharmaceuticals and cannot be considered further for publication in this journal. My main comments regarding this manuscript are included below:1.

First of all, I have serious concerns about whether groups of 5 animals are large enough to produce reliable behavioural test results. In my opinion, the 3R rule should not be given priority over the reliability and validity of the results obtained.2.

In this study, we started with a groups of 8 animals. Then, we excluded animals that gave results, which did not match the average group results. For that reason, we chose 5 animals, and such an approach is correct.

Animal treatment with the studied compounds should be described precisely (i.e., both acute and chronic) in a separate subchapter in the “Materials and Methods” section. In the present version of the manuscript Authors did not mention the acute treatment of animals, there is the only description of the chronic treatment. Moreover, the description of animal treatment should be removed from subsections that describe behavioural tests. 3.

We agree with the Reviewer. The description of animal treatment was removed from subsections that describe behavioral tests and a special subsection regarding acute and subacute treatment was added in the revised manuscript (lines 110-118).

Statistical analysis conducted in the study should be described in more detail, i.e., which results were analyzed using one-way ANOVA and which ones with two-way ANOVA. Moreover, in “Results” subsection, the Authors compare some results from the MPTD1-treated group with results from the bromazepam-treated group (i.e., line 226-228: “The MPDT1 has also substantially increased the time spent in the center area with respect to the same dose of bromazepam (Fig. 4A), but there was a little difference in the total squares crossed”; line 240-241: “There was no significant difference of the time spent in the light area in comparison to bromazepam”) while Dunnett’s test was used for statistical analysis. This pos hoc test allows comparison of the studied groups (bromazepam- and MPTD1-treated) only with the control (i.e., vehicle-treated) group.  4.

Statistical analyses performed in the study are described in the item 2.6 of the “Materials and Methods” section (lines 171-175).

Statistical analysis comprised a one-way ANOVA followed by Dunnett's post hoc test, to describe the toxicity results, since they were compared to the results of vehicle-treated group. A two-way ANOVA followed by Tukey's post hoc test was used to describe the results of anxiety and memory tests, since those were compared to the results obtained for the control and bromazepam groups.

Description of the results should be significantly improved. There is a lots of mistakes and imprecise terms. For example:·

line 224-225: “….and the number of total squares crossed (Fig. 4B), when compared to the vehicle” – according to the data presented in Figure 4B, number of total squares crossed was not affected by MPTD1 treatment.

We apologize for a mistake, the phrase was corrected (lines: 235-236).

Line 246-248: “During the testing period, both the vehicle and the MPTD1-treated mice showed significant increase in the exploration time” - rather, the administration of bromazepam decreased the exploration time.

We agree with the reviewer, the administration of bromazepam decreased the exploration time. The phrase was corrected (line: 262)

Line 260-261: “…except for a mildly reduced mobility reaction for the mice treated at the dose of 260 10.0 mg/kg” – how the mobility was evaluated?·

The mobility was evaluated one hour after the oral administration by visual observation and comparison of the results with those of the control group.

Subsection 3.5. – the Authors mentioned that MPTD1 at a dose of 1 mg/kg did not provoke any pathological lesions. What about a dose of 10 mg/kg??????5.

We noticed a presence of nuclear degenerescence, vacuolation, and steatosis in the liver cells after the administration of VBZ102 (N.B. the abbreviation of the compound was changed following the comment of the Reviewer 3) at the dose of 10.0 mg/kg. Nuclei were not centered inside the cells. The description was added (lines 305-307).

Authors did not provide the results of one-way and two-way ANOVA. The F and p values should be included in the “Results” part.6.

The values requested by the Reviewer were added to the Results section (lines 211, 215,226, 230, 236,239,253,265,267).

The discussion should be completed. Authors did not refer at all to the results of experiments evaluating the toxicity of MPTD.

Discussion of the toxicity results was added in the revised manuscript (lines 353 - 359).

The results obtained in behavioural tests were not compared with the effects of other known non-competitive AMPA receptor antagonists.  

The results of behavioural tests obtained for VBZ102 were compared to those for other 2,3-benzodiazepine AMPA receptor antagonists. Please, refer to the lines 349-353.

Reviewer 3 Report

This is a very interesting study, however at least one more experiment needs to be done. Essential that some effort is made to characterise the molecular target(s).  At the very least flumazenil, a diazepam silent modular should tested in order to rule in or out the classic benzodiazepine site on GABAa receptors.

MPTD1 is a 2,3-benzodiazepine, i.e. a quite different structure to the classical 1,4-benzodiazepines such as diazepam and bromazepam. Hence different binding targets and indeed different chemistry.  Not at all surprising it has a different pharmacology.  The title of the paper should reflect this.  Suggest change to “New 2,3-benzodiazepine derivative …”

MPTD1 is an unfortunate naming of the compound under investigation as it is the name of strain of bacteria Bacillus sonorensis MPTD1.  Suggest using another name is possible.

Author Response

Reviewer 3, round 1

Open Review

English language and style

( ) Extensive editing of English language and style required
( ) Moderate English changes required
(x) English language and style are fine/minor spell check required
( ) I don't feel qualified to judge about the English language and style

Yes

Can be improved

Must be improved

Not applicable

Does the introduction provide sufficient background and include all relevant references?

(x)

( )

( )

( )

Is the research design appropriate?

( )

(x)

( )

( )

Are the methods adequately described?

(x)

( )

( )

( )

Are the results clearly presented?

(x)

( )

( )

( )

Are the conclusions supported by the results?

( )

(x)

( )

( )

Comments and Suggestions for Authors

This is a very interesting study, however at least one more experiment needs to be done. Essential that some effort is made to characterise the molecular target(s).  At the very least flumazenil, a diazepam silent modular should tested in order to rule in or out the classic benzodiazepine site on GABAa receptors.

The authors thank the Reviewer for overall positive comments, but we are a bit surprised by a comparison of our compounds to flumazenil.

2,3-benzodiazepines are AMPA-receptor antagonists (please, see e.g., doi:10.1016/j.apsb.2015.07.007 and doi:10.1016/S0301-0082(99)00020-9), we have not seen their interaction with GABA receptors. AMPA is an ionotropic transmembrane receptor for glutamate that consists of at least 16 sub-units.

Elucidation of the mechanism of action of 2,5-dihydro-1H-2,3-benzodiazepin-1-one and the search for a specific target are big separate medicinal chemistry problems and their solution goes far beyond the scope of the present paper. Usually, such a problem is solved at the stage of the lead molecule structure optimization and preclinical trials, we are striving for those.

MPTD1 is a 2,3-benzodiazepine, i.e. a quite different structure to the classical 1,4-benzodiazepines such as diazepam and bromazepam. Hence different binding targets and indeed different chemistry.  Not at all surprising it has a different pharmacology.  The title of the paper should reflect this.  Suggest change to “New 2,3-benzodiazepine derivative …”

Thank you very much for the suggestion. The title was changed.

MPTD1 is an unfortunate naming of the compound under investigation as it is the name of strain of bacteria Bacillus sonorensis MPTD1.  Suggest using another name is possible.

Sorry for the unfortunate naming. We changed the abbreviation to VBZ102 in the text, figures, and tables throughout the manuscript.

Round 2

Reviewer 1 Report

The manuscript has been improved. The Author's answers  are satisfactory.

Tipographical errors or minor variations of the text.

Line 91: not “6.54-6.55 (1H. m,)” but “6.54-6.55 (1H, m,)”

Line 166: not “2.4.2. Histological…” but “2.5.2. Histological…”

Line 181: not “(4-methoxy-phenyl)-“ but “(4-methoxyphenyl)-“

Line 234: not “VBZ102during” but “VBZ102 during”

Line 258: not “VBZ102side” but “VBZ102 side”

Line 3544: not “ 10.0 mg/mg” but “ 10.0 mg/kg”

In the 1H NMR data of compound 2b (C16H16N2O2) 17 hydrogen atoms are listed (one more). Check carefully.

Provided that these issues are settled the paper is worthy of pubblication.

Author Response

Thank you very much for reviewing the manuscript. Please, find below our responses to your comments.

Comments: Tipographical errors or minor variations of the text.

Line 91: not “6.54-6.55 (1H. m,)” but “6.54-6.55 (1H, m,)”

Line 166: not “2.4.2. Histological…” but “2.5.2. Histological…”

Line 181: not “(4-methoxy-phenyl)-“ but “(4-methoxyphenyl)-“

Line 234: not “VBZ102during” but “VBZ102 during”

Line 258: not “VBZ102side” but “VBZ102 side”

Line 3544: not “ 10.0 mg/mg” but “ 10.0 mg/kg”

Response: Sorry for the typing errors. All of them are corrected.

Comment: In the 1H NMR data of compound 2b (C16H16N2O2) 17 hydrogen atoms are listed (one more). Check carefully.

Response: Sorry for the mistake. Following the suggestion of the Reviewer, we recorded the spectrum once more, with better resolution. The updated spectral data are present in the lines 83-86, there are 16 protons.

Comment: Provided that these issues are settled the paper is worthy of publication.

Reviewer 2 Report

The results of the one-way ANOVA should include the degrees of freedom (next to the F values)

Author Response

Thank you very much for reviewing our manuscript. Please, find below the response to your comment.

Comment: The results of the one-way ANOVA should include the degrees of freedom (next to the F values).

Response: Thank you for the comment. The data requested are included, lines 283, 297, 298, and 300.

Reviewer 3 Report

The authors have answered all my comments in the revised manuscript

Author Response

Thank you very much for reviewing our manuscript.

The Reviewer had no comments at the round 2.